# Development of a Novel Anti-EpCAM Monoclonal Antibody for Various Applications

**DOI:** 10.3390/antib11020041

**Published:** 2022-06-08

**Authors:** Guanjie Li, Hiroyuki Suzuki, Teizo Asano, Tomohiro Tanaka, Hiroyoshi Suzuki, Mika K. Kaneko, Yukinari Kato

**Affiliations:** 1Department of Molecular Pharmacology, Tohoku University Graduate School of Medicine, 2-1 Seiryo-machi, Aoba-ku, Sendai 980-8575, Japan; kanketsu@med.tohoku.ac.jp; 2Department of Antibody Drug Development, Tohoku University Graduate School of Medicine, 2-1 Seiryo-machi, Aoba-ku, Sendai 980-8575, Japan; teizo.asano.a7@tohoku.ac.jp (T.A.); tomohiro.tanaka.b5@tohoku.ac.jp (T.T.); k.mika@med.tohoku.ac.jp (M.K.K.); 3Department of Pathology and Laboratory Medicine, Sendai Medical Center, 2-11-12, Miyagino, Miyagino-ku, Sendai 983-0045, Japan; hirosuzsnh@gmail.com

**Keywords:** EpCAM, monoclonal antibody, recombinant antibody, colorectal carcinoma

## Abstract

The epithelial cell adhesion molecule (EpCAM) is a cell surface glycoprotein, which is widely expressed on normal and cancer cells. EpCAM is involved in cell adhesion, proliferation, survival, stemness, and tumorigenesis. Therefore, EpCAM is thought to be a promising target for cancer diagnosis and therapy. In this study, we established anti-EpCAM monoclonal antibodies (mAbs) using the Cell-Based Immunization and Screening (CBIS) method. We characterized them using flow cytometry, Western blotting, and immunohistochemistry. One of the established recombinant anti-EpCAM mAbs, recEpMab-37 (mouse IgG_1_, kappa), reacted with EpCAM-overexpressed Chinese hamster ovary-K1 cells (CHO/EpCAM) or a colorectal carcinoma cell line (Caco-2). In contrast, recEpMab-37 did not react with EpCAM-knocked out Caco-2 cells. The *K*_D_ of recEpMab-37 for CHO/EpCAM and Caco-2 was 2.0 × 10^−8^ M and 3.2 × 10^−8^ M, respectively. We observed that EpCAM amino acids between 144 to 164 are involved in recEpMab-37 binding. In Western blot analysis, recEpMab-37 detected the EpCAM of CHO/EpCAM and Caco-2 cells. Furthermore, recEpMab-37 could stain formalin-fixed paraffin-embedded colorectal carcinoma tissues by immunohistochemistry. Taken together, recEpMab-37, established by the CBIS method, is useful for detecting EpCAM in various applications.

## 1. Introduction

The epithelial cell adhesion molecule (EpCAM) is an approximately 40 kilodalton (kDa), type I transmembrane glycoprotein, which is expressed on the basolateral membrane of most epithelial cells [1]. EpCAM is a unique cell adhesion molecule that does not resemble other classical adhesion molecules (e.g., cadherins, selectins, and integrins) in either structure or function. EpCAM forms homophilic and intercellular adhesion, which plays critical roles in the maintenance of the epithelial integrity [2]. EpCAM also serves as a signaling molecule. Upon cleavage of EpCAM at the cell membrane, the intracellular domain of EpCAM serves as a transcriptional cofactor with β-catenin and regulates the transcription of target genes involved with cell proliferation, survival, and the maintenance of stem cell properties [3].

In addition to its role in normal epithelial tissues, EpCAM plays an important role in tumor development, and was the first identified human tumor-associated antigen [4]. Studies have shown that EpCAM is overexpressed in many epithelial cancers including pancreas, colorectal, and prostate cancers [5]. Due to high expression and antigenicity, EpCAM has been considered as a target of monoclonal antibody (mAb) therapies in colorectal cancer [6,7]. More recently, a bispecific EpCAM/CD3-antibody, Catumaxomab, serves as the trifunctional antibody. Catumaxomab binds to EpCAM-overexpressed tumors and recruits T cells to tumor cells, which promote cytotoxic synapse formation and tumor cell lysis. Furthermore, through the Fc domain, Catumaxomab also recruits natural killer cells and macrophages and enhances antibody-dependent cellular cytotoxicity (ADCC) [8,9]. Several clinical trials have demonstrated promising outcomes [10,11,12], and Catumaxomab has been approved by the European Union for the treatment of patients with malignant ascites in EpCAM-positive carcinomas [13].

Moreover, anti-EpCAM mAbs are most widely used to collect circulating tumor cells (CTCs), which are useful indicators of micro-metastasis and provide important prognostic information to determine the response to therapeutic interventions [14]. The clinical importance of CTCs was confirmed by the U.S. Food and Drug Administration (FDA) approval of CellSearch, a platform that could extract EpCAM-positive, CD45-negative cells from whole blood samples [15]. Therefore, the establishment of anti-EpCAM mAbs is important to improve the efficacy of the abovementioned applications and to develop novel modalities for tumor therapy. In this study, we developed anti-EpCAM mAbs using the Cell-Based Immunization and Screening (CBIS) method [16,17,18,19,20,21,22,23,24,25,26,27,28,29,30,31], and evaluated it for various applications, including flow cytometry, Western blotting, and immunohistochemical analyses.

## 2. Materials and Methods

### 2.1. Plasmids

The Genome Network Project clone IRAK021G03 (EpCAM) was provided by the RIKEN BioResource Research Center through the National BioResource Project of the MEXT and AMED agencies of Japan. EpCAM cDNA plus a C-terminal PA tag that is recognized by the anti-PA tag mAb (NZ-1), was subcloned into a pCAG-Ble vector (FUJIFILM Wako Pure Chemical Corporation, Osaka, Japan). N-terminal PA-tagged EpCAM deletion mutants (dN44, dN64, dN84, dN104, dN124, dN144, dN164, dN184, dN204, dN224, and dN244) were produced using a HotStar HiFidelity Polymerase Kit (Qiagen Inc., Hilden, Germany), and subcloned into the pCAG-Ble vector.

### 2.2. Cell Lines

P3X63Ag8U.1 (P3U1), CHO-K1, and Caco-2 (human colorectal carcinoma) cells were obtained from the American Type Culture Collection (ATCC, Manassas, VA, USA). CHO/EpCAM and CHO/EpCAM deletion mutants were established by transfecting pCAG/EpCAM vectors into CHO-K1 cells using the Neon Transfection System (Thermo Fisher Scientific, Inc., Waltham, MA, USA) as described previously [32]. EpCAM-knocked out Caco-2 (BINDS-16) was established as described previously [33], and the related cell lines are available from Tohoku University Cell Bank (Kato’s Lab; http://www.med-tohoku-antibody.com/topics/001_paper_cell.htm (accessed on 5 June 2022)). P3U1, CHO-K1, and CHO/EpCAM were cultured in RPMI 1640 media (Nacalai Tesque, Inc., Kyoto, Japan). Caco-2 was cultured in Dulbecco’s modified Eagle’s medium (DMEM; Nacalai Tesque, Inc., Kyoto, Japan). The media were supplemented with 10% heat-inactivated fetal bovine serum (FBS; Thermo Fisher Scientific Inc., Waltham, MA, USA), 100 U/mL of penicillin, 100 μg/mL streptomycin, and 0.25 μg/mL amphotericin B (Nacalai Tesque, Inc.). Cells were incubated at 37 °C in a humidified atmosphere containing 5% CO_2_.

### 2.3. Animals

A BALB/c mouse was purchased from CLEA Japan (Tokyo, Japan). The animal was housed under specific pathogen-free conditions. All animal experiments were conducted following the relevant guidelines and regulations to minimize animal suffering and distress in the laboratory. Animal experiments were approved by the Animal Care and Use Committee of Tohoku University (Permit number: 2019NiA-001). The mouse was monitored daily for health during the full four weeks duration of the experiment. A reduction of more than 25% of the total body weight was defined as a humane endpoint. The mouse was euthanized by cervical dislocation, and death was verified by respiratory and cardiac arrest.

### 2.4. Hybridoma Production

We used the CBIS method to develop mAbs against EpCAM. Briefly, one BALB/c mouse was intraperitoneally (i.p.) immunized with CHO/EpCAM cells (1 × 10^8^) together with Imject Alum adjuvant (Thermo Fisher Scientific Inc.). The procedure included three additional immunizations, followed by a final booster injection-administered i.p. two days before spleen cell harvesting. Spleen cells were then fused with P3U1 cells using PEG1500 (Roche Diagnostics, Indianapolis, IN, USA). Hybridomas were grown in RPMI 1640, supplemented with 10% FBS, 100 U/mL of penicillin, 100 μg/mL streptomycin, and 0.25 μg/mL amphotericin B. Hypoxanthine, aminopterin, and thymidine (HAT; Thermo Fisher Scientific Inc.) were also added for hybridoma selection. Culture supernatants were screened using flow cytometry.

### 2.5. Production of the Recombinant Antibody

Variable (V_H_) and constant (C_H_) regions of heavy chain cDNAs of EpMab-37 were subcloned into the pCAG-Neo vector (FUJIFILM Wako Pure Chemical Corporation, Osaka, Japan) along with variable (V_L_) and constant (C_L_) regions of light chain cDNAs of EpMab-37 into the pCAG-Ble vector (FUJIFILM Wako Pure Chemical Corporation) to produce recombinant EpMab-37 (recEpMab-37). EpMab-37 vectors were transfected into ExpiCHO-S cells using the ExpiCHO Expression System (Thermo Fisher Scientific Inc.). recEpMab-37 was purified using Ab-Capcher (ProteNova, Kagawa, Japan).

### 2.6. Flow Cytometry

CHO/EpCAM, CHO-K1, Caco-2, and BINDS-16 cells were harvested after brief exposure to 0.25% trypsin in 1 mM ethylenediaminetetraacetic acid (EDTA; Nacalai Tesque, Inc.). After washing with 0.1% bovine serum albumin (BSA) in phosphate-buffered saline (PBS), cells were treated with recEpMab-37 for 30 min at 4 °C, and then with Alexa Fluor 488-conjugated anti-mouse IgG (1:1000; Cell Signaling Technology, Inc., Danvers, MA, USA). Fluorescence data were collected using SA3800 Cell Analyzer (Sony Corp., Tokyo, Japan).

### 2.7. Determination of the Binding Affinity

CHO/EpCAM and Caco-2 cells were suspended in 100 μL of serially diluted recEpMab-37 (6 ng/mL–100 μg/mL), followed by the addition of Alexa Fluor 488-conjugated anti-mouse IgG (1:200). Fluorescence data were collected using BD FACSLyric (BD Biosciences, Franklin Lakes, NJ). The dissociation constant (*K*_D_) was calculated by fitting binding isotherms to built-in, one-site binding models in GraphPad Prism 8 (GraphPad Software, Inc., La Jolla, CA, USA).

### 2.8. Western Blot Analysis

Cell lysates of CHO/EpCAM, CHO-K1, Caco-2, and BINDS-16 cells were boiled in sodium dodecyl sulfate sample buffer (Nacalai Tesque, Inc.). The samples were then electrophoresed on 5–20% polyacrylamide gels (Nacalai Tesque, Inc.) and transferred onto polyvinylidene difluoride (PVDF) membranes (Merck KGaA, Darmstadt, Germany). After blocking with 4% skim milk (Nacalai Tesque, Inc.) for 1 h, the membrane was incubated with recEpMab-37 (1 μg/mL) or anti-β-actin (1 μg/mL) for 1 h, followed by incubation with HRP-conjugated anti-mouse immunoglobulins (Agilent Technologies, Inc., Santa Clara, CA, USA) at a 1:2000 dilution for 1 h. The membrane was developed using the ImmunoStar LD Chemiluminescence Reagent (FUJIFILM Wako Pure Chemical Corporation) and a Sayaca-Imager (DRC Co., Ltd., Tokyo, Japan). All Western blot procedures were performed at room temperature.

### 2.9. Immunohistochemical Analysis

Histologic sections that were 4 μm thick of colorectal carcinoma tissue array (Catalog number: CO243b; US Biomax Inc., Rockville, MD, USA) were autoclaved in citrate buffer (pH 6.0; Agilent Technologies Inc.) for 20 min. Sections were then incubated with 1 μg/mL of recEpMab-37 for 1 h at room temperature and treated using an Envision+ kit (Agilent Technologies, Inc.) for 30 min. The color was developed using 3,3′-diaminobenzidine tetrahydrochloride (DAB; Agilent Technologies, Inc.) for 2 min, and sections were then counterstained with hematoxylin (FUJIFILM Wako Pure Chemical Corporation).

## 3. Results

### 3.1. Flow Cytometric Analysis of recEpMab-37 to EpCAM-Expressing Cells

In this study, EpMab-37 was established by immunizing one mouse with CHO/EpCAM cells (Figure 1). We confirmed the reactivity of EpMab-37 to CHO/EpCAM cells, cloned the cDNAs of EpMab-37, and produced the recombinant mAb, recEpMab-37.

We first confirmed the reactivity of recEpMab-37 against CHO/EpCAM by flow cytometry. As shown in Figure 2A, recEpMab-37 recognized CHO/EpCAM cells in a dose-dependent manner, but not CHO-K1 cells (Figure 2B). We next examined whether recEpMab-37 could recognize endogenous EpCAM in colorectal carcinoma Caco-2 cells. recEpMab-37 reacted with Caco-2 cells in a dose-dependent manner (Figure 2C). However, recEpMab-37 showed no reaction with EpCAM-knocked out Caco-2 (BINDS-16) cells (Figure 2D).

Then, we determined the apparent binding affinity of recEpMab-37 with CHO/EpCAM and Caco-2 using flow cytometry. The *K*_D_ of recEpMab-37 for CHO/EpCAM and Caco-2 was 2.0 × 10^−8^ M and 3.2 × 10^−8^ M, respectively, indicating that recEpMab-37 possesses the moderate affinity for EpCAM-expressing cells (Figure 3).

We next generated eleven N-terminal PA tagged EpCAM deletion mutants transfected CHO-K1 cells, namely dN44 (corresponding to 44–314 amino acids [aa]), dN64 (corresponding to 64–314 aa), dN84 (corresponding to 84–314 aa), dN104 (corresponding to 104–314 aa), dN124 (corresponding to 124–314 aa), dN144 (corresponding to 144–314 aa), dN164, (corresponding to 164–314aa), dN184 (corresponding to 184–314 aa), dN204, (corresponding to 204–314 aa), dN224, (corresponding to 224–314 aa), and dN244, (corresponding to 244–314 aa). All deletion mutants of EpCAM containing the N-terminal PA tag were detected by NZ-1 (an anti-PA tag mAb), indicating the expression level of each construct to be similar (Figure 4A). Although recEpMab-37 detected dN44, dN64, dN84, dN104, dN124, and dN144, it did not react with dN164, dN184, dN204, dN224, and dN244 (Figure 4B), suggesting that the recEpMab-37 epitope exists between 144 and 164 aa of EpCAM.

### 3.2. Western Blot Analysis

Western blotting was performed to further assess the sensitivity of recEpMab-37. Lysates of CHO-K1 and CHO/EpCAM cells were probed. As shown in Figure 5, recEpMab-37 detected EpCAM as 35~40 kDa double bands. However, recEpMab-37 did not detect any bands from lysates of CHO-K1 cells. These results indicate that recEpMab-37 specifically detects exogenous EpCAM. Furthermore, recEpMab-37 could detect endogenous EpCAM as a 35-kDa band from lysates of Caco-2 cells. In contrast, recEpMab-37 could not detect any bands from lysates of EpCAM-knocked out Caco-2 (BINDS-16) cells. These results suggest that recEpMab-37 specifically recognizes endogenous EpCAM in the colorectal carcinoma cell line.

### 3.3. Immunohistochemical Analysis Using recEpMab-37 against Colorectal Carcinoma Tissues

We investigated whether recEpMab-37 is applicable for immunohistochemical analyses using formalin-fixed paraffin-embedded (FFPE) colorectal carcinoma sections. recEpMab-37 strongly stained colorectal carcinoma cells in a colorectal carcinoma section (Figure 6A,B), but not stromal tissues (Figure 6A). A cytoplasmic staining pattern in carcinoma cells was observed (Figure 6B). Hematoxylin and eosin (HE) staining was performed using the serial sections (Figure 6C,D). In contrast, recEpMab-37 showed faint staining in the surface of normal colorectal epithelium (Figure 6E,F). HE staining was also performed using the serial sections (Figure 6G,H). These results indicate that recEpMab-37 is useful for the immunohistochemical analysis of FFPE tumor sections.

## 4. Discussion

In this study, we first developed EpMab-37 using the CBIS method and next produced its recombinant mAb (recEpMab-37) (Figure 1). Then, we investigated the usefulness of recEpMab-37 for flow cytometry (Figure 2), Western blotting (Figure 5), and immunohistochemistry (Figure 6). Since recEpMab-37 does not detect any bands in EpCAM-knocked out Caco-2 cells, recEpMab-37 exhibited high specificity to EpCAM. To determine the epitope of recEpMab-37, we first performed an enzyme-linked immunosorbent assay using linear peptides in the EpCAM extracellular domain. However, EpMab-37 never recognized any peptides including 144 to 163 aa of EpCAM (Appendix A). Next, we confirmed that the 144 to 164 aa of EpCAM are involved in recEpMab-37 binding using the deletion mutant-expressed CHO-K1 cells (Figure 4). Because EpMab-37 was established by the immunization of EpCAM-overexpressed CHO-K1 cells, EpMab-37 could recognize the native and conformational epitope. However, we cannot rule out the possibility that the deletion of EpCAM affects the native conformation and collapse the recEpMab-37 epitope. We have developed the Arg, Ile, Glu, Asp, and Leu (RIEDL)-insertion for epitope mapping (REMAP) and His-tag insertion for the epitope mapping (HisMAP) method to determine the conformational epitopes. We successfully determined the epitopes of anti-EGFR mAbs (EMab-51 and EMab-134) [34,35] and anti-CD44 mAbs (C_44_Mab-5 and C_44_Mab-46) [36,37] by the REMAP method, and anti-CD20 mAbs (C_20_Mab-11 and C_20_Mab-60) using the HisMAP method [38,39]. The determination of EpMab-37 epitope may provide useful information for future applications.

EpCAM plays critical roles in tumorigenesis to promote tumor cell proliferation, survival, and tumor-initiating potential [2]. EpCAM has been reported to be overexpressed in several tumors [40,41,42,43,44], which correlates with poor prognosis, therapeutic failure, and early tumor recurrence [43]. Therefore, EpCAM has been considered as a promising therapeutic target. A humanized scFv against EpCAM, fused to Pseudomonas exotoxin A, Oportuzumab monatox (Vicinium, VB4-845), was developed [3]. After Vicinium targets EpCAM on tumor cells, it is internalized, receives the cleavage of the linker, releases the payload exotoxin, and exhibits cytotoxicity. Vicinium was granted the FDA fast track designation for the treatment of bacillus Calmette-Guérin (BCG)-unresponsive non-muscle invasive bladder cancer (NMIBC) [45]. Vicinium was evaluated in the single-arm Phase 3 VISTA study (NCT02449239) for the treatment of patients with high-grade NMIBC in situ and high-grade papillary disease previously treated with BCG. However, FDA did not approve Vicinium for BCG-unresponsive NMIBC in August 2021 [46].

As mentioned above, clinical applications of anti-EpCAM mAbs are still limited since anti-EpCAM mAbs might generate a degree of toxicity against normal tissues. We previously established cancer-specific mAbs (CasMabs) targeting podoplanin [47,48,49,50] and podocalyxin [51], which are expressed on many tumors [52,53,54,55]. It is worthwhile to develop cancer-specific anti-EpCAM mAbs using the CasMab method. We previously converted a mouse IgG_1_ subclass into a mouse IgG_2a_, and produced a defucosylated version which exhibited high ADCC activities in vitro, and significantly reduced tumor growth [56]. Therefore, the production of a class-switched and defucosylated version of EpMab-37 is warranted to evaluate in vivo antitumor activity.

## Figures and Tables

**Figure 1 antibodies-11-00041-f001:**
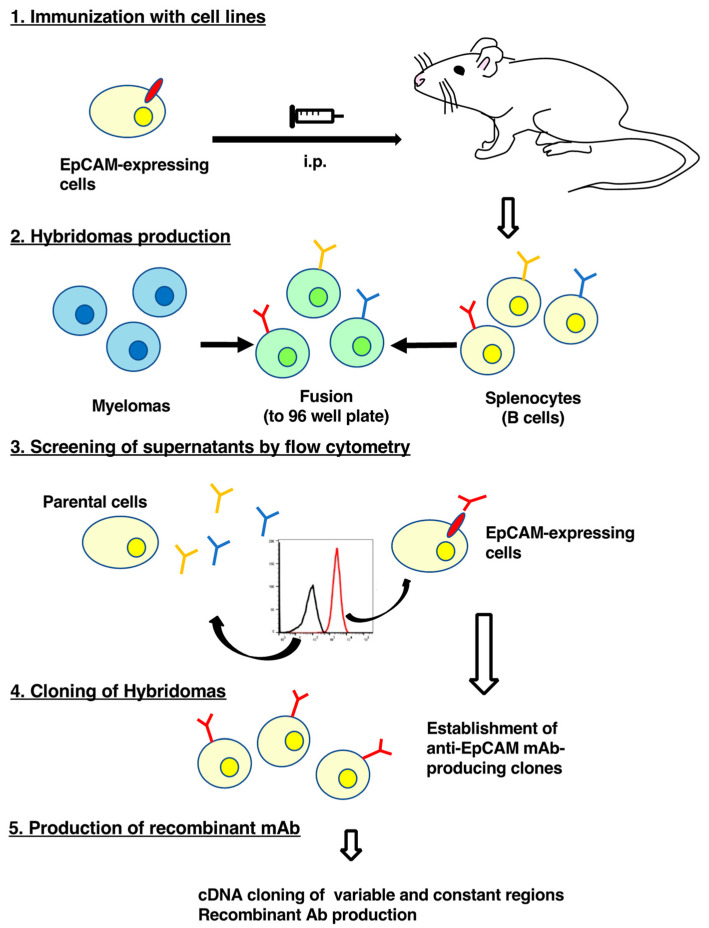
A schematic procedure of anti-EpCAM mAbs production. A mouse was intraperitoneally immunized with the CHO/EpCAM cells. The screening was then conducted by flow cytometry using parental cells and EpCAM-overexpressed CHO-K1 cells. Variable and constant regions cDNAs were cloned from an established clone, and the recombinant Ab was produced.

**Figure 2 antibodies-11-00041-f002:**
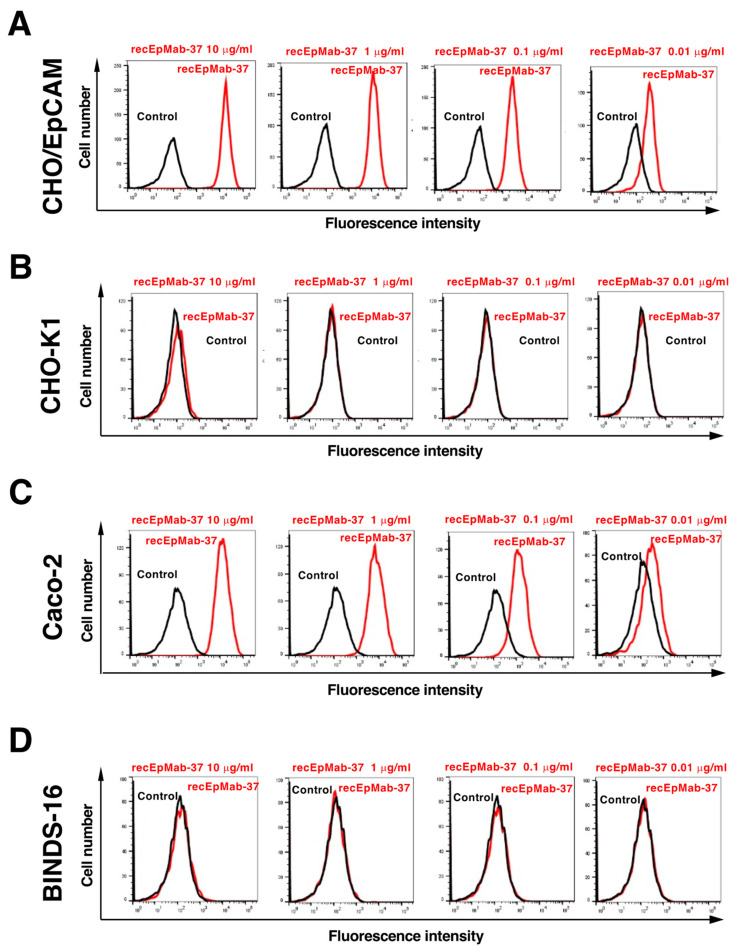
Flow cytometry of EpCAM expressing cells using recEpMab-37. CHO/EpCAM (**A**), CHO-K1 (**B**), Caco-2 (**C**), and BINDS-16 (EpCAM-knocked out Caco-2) (**D**) cells were treated with 0.01–10 µg/mL of recEpMab-37, followed by treatment with Alexa Fluor 488-conjugated anti-mouse IgG. Black line represents the negative control.

**Figure 3 antibodies-11-00041-f003:**
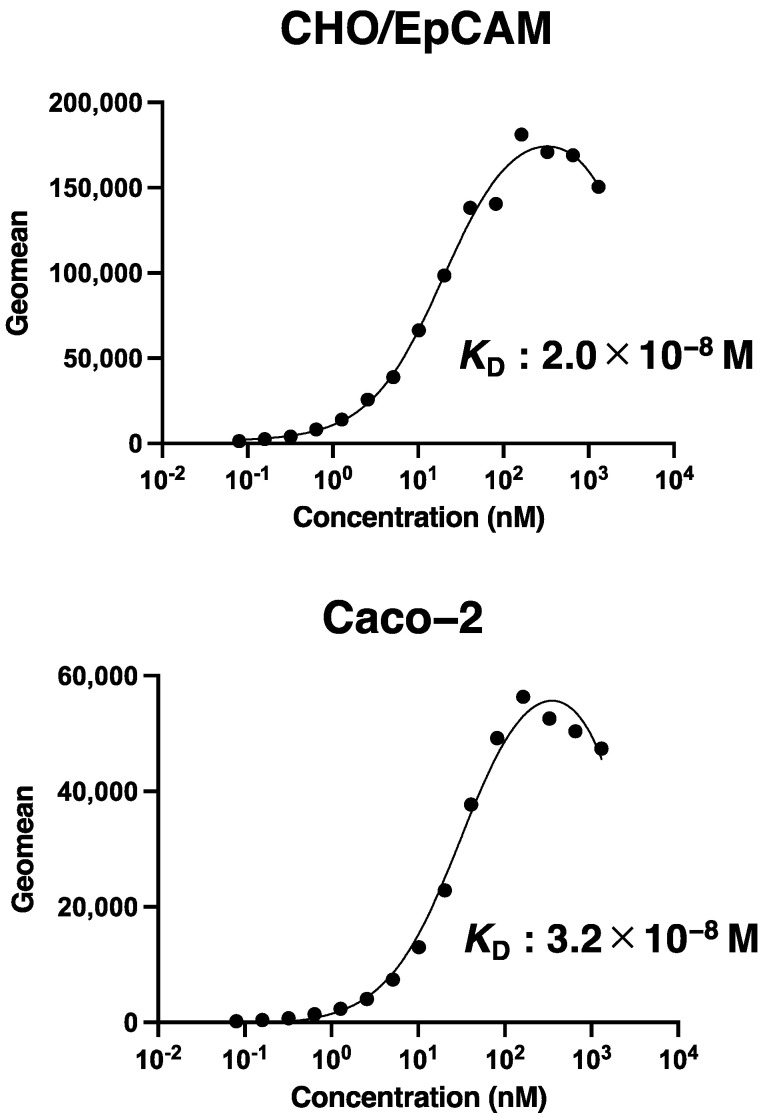
The determination of the binding affinity of recEpMab-37. CHO/EpCAM and Caco-2 cells were suspended in 100 µL serially diluted recEpMab-37 (6 ng/mL to 100 µg/mL). Then, cells were treated with Alexa Fluor 488-conjugated anti-mouse IgG. Fluorescence data were subsequently collected using a BD FACSLyric, following the calculation of the dissociation constant (*K*_D_) by GraphPad PRISM 8.

**Figure 4 antibodies-11-00041-f004:**
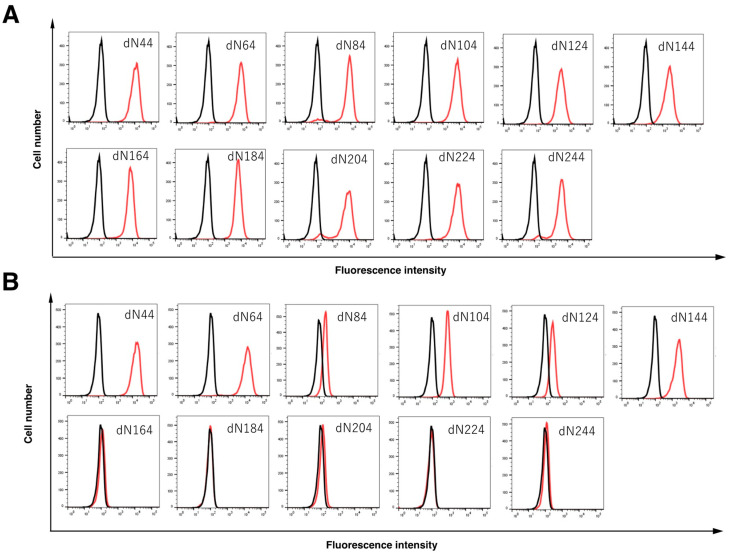
Epitope mapping of recEpMab-37 using deletion mutants of EpCAM. CHO-K1 expressed EpCAM deletion mutants were analyzed using flow cytometry. The EpCAM mutants-transfected cells were incubated with NZ-1 (anti-PA tag; red line, (**A**)), recEpMab-37 (anti-EpCAM mAb; red line, (**B**)), or buffer control (black line, (**A**,**B**)) for 30 min at 4 °C, followed by secondary antibodies.

**Figure 5 antibodies-11-00041-f005:**
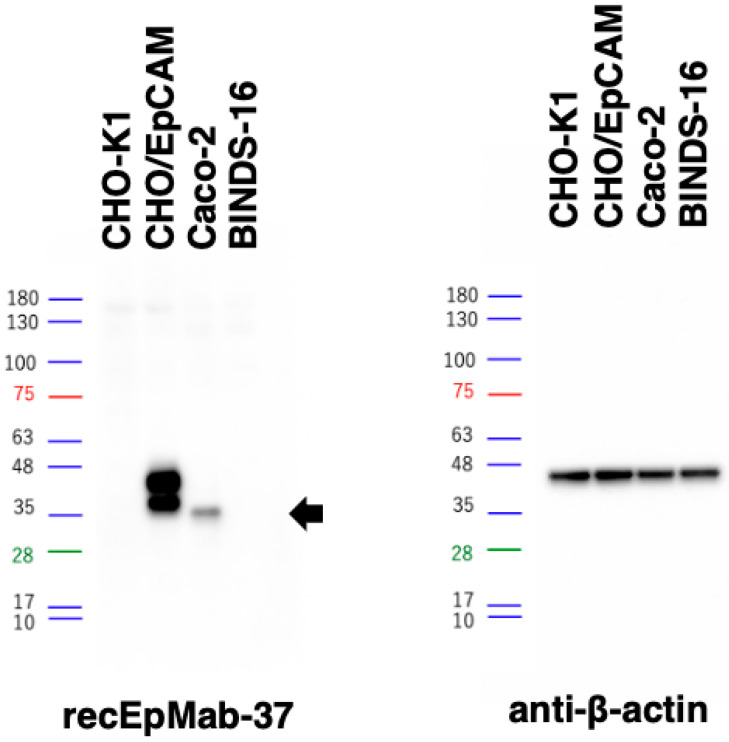
Western blotting by recEpMab-37. Cell lysates of CHO-K1, CHO/EpCAM, Caco-2, and BINDS-16 (10 µg) were electrophoresed and transferred onto polyvinylidene fluoride membranes. The membranes were incubated with 1 µg/mL of recEpMab-37 and 1 µg/mL of anti-β-actin and subsequently with peroxidase-conjugated anti-mouse immunoglobulins. Black arrows indicate the predicted size of EpCAM (~35 kDa).

**Figure 6 antibodies-11-00041-f006:**
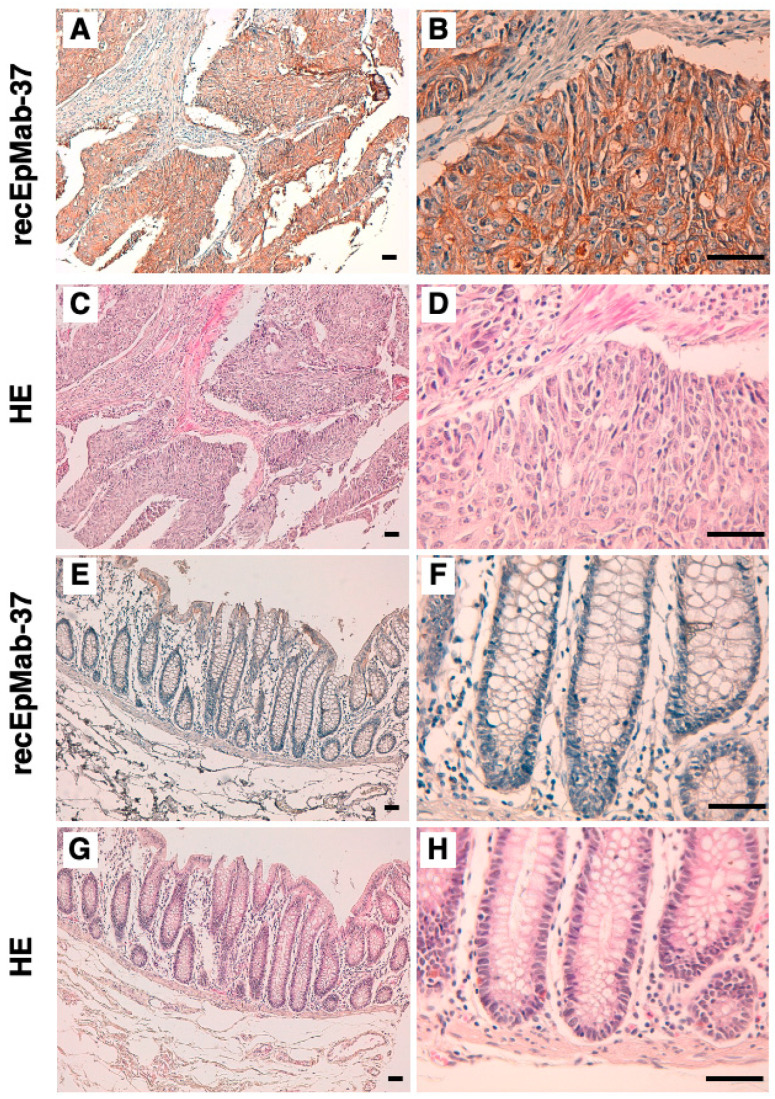
Immunohistochemical analysis using recEpMab-37 against colorectal carcinoma tissues. (**A**,**B**,**E**,**F**) After antigen retrieval, sections were incubated with 1 µg/mL of recEpMab-37 followed by treatment with the Envision+ kit. Color was developed using DAB, and sections were counterstained with hematoxylin. Cytoplasmic brown staining indicates the EpCAM positive cells. (**C**,**D**,**G**,**H**) Hematoxylin and eosin (HE) staining. Scale bar = 100 µm.

## Data Availability

The data presented in this study are available in the article and Appendix A.

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
