# Peer review of "Development of a Novel Anti-EpCAM Monoclonal Antibody for Various Applications"

_2073-4468, 2022, doi:10.3390/antib11020041_

Round 1

Reviewer 1 Report

It would aid the reader if the authors  provided additional support and rationale for the “Determination of binding affinity” outlined the “Methods and Material” section and corresponding Results section.

Specifically, provide if, possible the number of epCAM “receptors”  found on the CHO/EpCAM and Cacao-2 cell lines. It is apparent from the  binding data outlined in Figure 3 that Caco-2 has three times more receptors. It is also seen from the Western Blot (Figure 4), that CHO cells express two forms of EpCAM. The authors should examine and discuss how these differing analytes will impact on their Kd calculations.

The authors state that on lines 157-158 “… recEpMab-37 possesses the moderate  affinity for epCAM expressing cells.   Please provide a reference and support for the term “moderate”. Is the epMab-37 moderate affinity  as compared to other epCAM antibodies found in the literature, their earlier reported EpMab-16 (references 32,33) or compared to monoclonal antibody affinities in general?

The authors should note a caveat in the results section for in their affinity models and Kd calculations. The authors are using the intact antibody and not normally used single hit the F(ab) fragment. Use of whole antibody will lead to cooperative receptor binding and may result in an apparent increase in the Kd value compared to that seen if using a single hit model and use of a F(ab) fragment.

The authors suggest future experiments wherein epMAB-37 epitope mapping may be performed. As a suggestion to the authors, this work may be hasten and simplified it the authors employed antibody competition experiments and use of commercially available antibodies for which the exact peptide epitope or epCAM epitope domain is reported.

Minor comment:

Please revise the use of “never” to “showed no reaction” (lines 153-154). However, recEpMab-37 never reacted with EpCAM-knocked out Caco-2 (BINDS-16) cells (Fig. 2D).

Please revise,  “In contrast, recEpMab-37 did not stain normal colorectal epithelium (Fig. 5E and 5F), (lines 197-198).” This reviewer’s close examination of normal colorectal epithelium  (Fig 5 panel E) does show a brown, albeit faint, coloring on the epithelium.

Reviewer 2 Report

The manuscript by Li et al. describes the generation and characterization of a mouse monoclonal antibody against human EpCAM. The antibody recEpMab-37 was produced by hybridoma from a mouse immunzed with EpCAM-producing cell line, and was able to detect EpCAM in Western blotting, flow cytometry, and immunohistochemical staining.

While the antibody described in the manuscript could detect human EpCAM with high specificity and moderate affinity, the authors did not perform any further studies regarding the antibody's functions. As of now recEpMab-37 is just another reagent-grade anti-EpCAM antibody, the likes of which anyone can buy from vendors. The studies proposed in the Discussion section (e.g., epitope mapping, cancer-specific targeting, or glycoengineering for enhanced effector function) would significantly improve the quality of the research. 

Reviewer 3 Report

This is a solid work, showing the isolation and characterization of an anti-EpCAM monoclonal antibody. 

My one main criticism, and one that I repeat with every review, is: sequences of the antibodies MUST be shown; if this work is in patent analysis, sequences will be protected (and published) anyway; if not, sequences should be released so others can also profit scientifically from this work.

My minor points, mostly concerning English, are here below:

Fig 1: “1. immunization OF cells lines” or “WITH cell lines”?

Fig 2: Flow cytometry OF EpCAM expressing cells

Fig 4: What is the reason 2 bands were detected on WB for the exogenous EpCAM?

Fig 5: useful to indicate with arrows an example of labelling, as this is not always obvious for people unfamiliar with IHC.

line 43: was the first identified a human tumor-associated antigen

line 44: EpCAM is overexpressed in the many epithelial cancers

line 47: a bispecific EpCAM/CD3-antibody, Catumaxomab, serveS

line 69: pCAG/EpCAM-PA: what is exactly this construct? From which source? It might be described on the references, but it’s very useful to repeat this information here.

Line 81: “A BALB/c mouse was purchased ...  The animal was housed”. If truly only 1 mouse was used, then on line 85: Mice was -> The mouse was

Line 212: recEpMab-37 exhibited the high specificity to EpCAM.

Reviewer 4 Report

This is an interesting and important study that describes development  of a mAb that can be used to target EpCam.

I have just a few comments/correction:

Line 37 should be: “in the maintenance of the”

Line 48: tumors

Line 56: provide

Line 89: Please explain: WHY was only one mouse used? Usually, one immunizes many mice to be sure to get good mAbs

Point 1.     Figure 1 should be : Immunization with cell lines

Line 148: immunized with the

Line 167: mutants in transfected 

Round 2

Reviewer 1 Report

Although it is the authors choice to include or exclude data. It would be helpful in future application of this, or other lab generated/reported anti-EpCam monoclonal antibodies that Kd and other pertinent information be included. The data may be useful in the design of future applications.

The author’s state: "Although we confirmed that the epitope of recEpMab-37 exists between 144 and 164 aa of EpCAM (Fig. 3), further studies are warranted to determine the detailed binding epitope of EpMab-37. We could not determine the epitope of EpMab-37 using conventional peptides alanine scanning between 144 and 164 aa of EpCAM (data not shown)" (Lines 231-234).
The authors have not confirmed that the epitope exists between AA 144-164. There are alternative and equally valid explanations as to why the antibody did not recognize the EpCAM deletion mutants. For example, the domain may not have folded properly or lacked the ability to interact with a distal region of the protein molecule such that mAb-37 could not bind. At best the authors should state that this region is involved in mAb-37 epitope recognition.
The authors should report the "data not shown" as supplemental data.
The authors could readily identify the mAb-37 epitope using peptide competition experiments. 
Minor comment: We also found that the recEpMab-37 epitope exists between 144 and 164 amino acids of EpCAM. (Lines 23-24)
Suggested revision: “We observed that EpCAM amino acids 144 to 164 are involved in recEpMab-37 binding.

Reviewer 2 Report

The authors provided additional data on epitope mapping, which can be useful when trying to further develop/optimize the molecule. Nonetheless, the manuscript still reports a reagent-grade antibody that can be used for flow cytometry, immunoblotting, and IHC. I failed to notice novelty in terms of target antigen, antibody generation technology, or functionality of the antibody reported here.
